# GMFCS Level-Specific Differences in Kinematics and Joint Moments of the Involved Side in Unilateral Cerebral Palsy

**DOI:** 10.3390/jcm11092556

**Published:** 2022-05-02

**Authors:** Stefanos Tsitlakidis, Nicholas A. Beckmann, Sebastian I. Wolf, Sébastien Hagmann, Tobias Renkawitz, Marco Götze

**Affiliations:** Clinic for Orthopedics, Heidelberg University Hospital, Schlierbacher Landstrasse 200a, 69118 Heidelberg, Germany; stefanos.tsitlakidis@med.uni-heidelberg.de (S.T.); nicholas.beckmann@med.uni-heidelberg.de (N.A.B.); sebastian.wolf@med.uni-heidelberg.de (S.I.W.); sebastien.hagmann@med.uni-heidelberg.de (S.H.); tobias.renkawitz@med.uni-heidelberg.de (T.R.)

**Keywords:** 3D-instrumented gait analysis, unilateral cerebral palsy, GMFCS level-specific differences, kinematic parameters, joint moments

## Abstract

A variety of gait pathologies is seen in cerebral palsy. Movement patterns between different levels of functional impairment may differ. The objective of this work was the evaluation of Gross Motor Function Classification System (GMFCS) level-specific movement disorders. A total of 89 individuals with unilateral cerebral palsy and no history of prior treatment were included and classified according to their functional impairment. GMFCS level-specific differences, kinematics and joint moments, exclusively of the involved side, were analyzed for all planes for all lower limb joints, including pelvic and trunk movements. GMFCS level I and level II individuals most relevantly showed equinus/reduced dorsiflexion moments, knee flexion/reduced knee extension moments, reduced hip extension moments with pronounced flexion, internal hip rotation and reduced hip abduction. Anterior pelvic tilt, obliquity and retraction were found. Individuals with GMFCS level II were characterized by an additional pronounced reduction in all extensor moments, pronounced rotational malalignment and reduced hip abduction. The most striking characteristics of GMFCS level II were excessive anterior pelvic/trunk tilt and excessive trunk obliquity. Pronounced reduction in extensor moments and excessive trunk lean are distinguishing features of GMFCS level II. These patients would benefit particularly from surgical treatment restoring pelvic symmetry and improving hip abductor leverage. Future studies exploring GMFCS level-specific compensation of the sound limb and GMFCS level-specific malalignment are of interest.

## 1. Introduction

Cerebral palsy (CP) is a heterogeneous and complex neuromuscular disorder that leads to different degrees of severity [1,2,3,4]. The extent of the secondary movement disorder, on the one hand, depends on the extent of the primary brain injury and its etiology, the extent of asymmetry and walking speed, and on the other hand, is positively influenced by plasticity and reorganization of neuronal networks and practice-induced improvements [5,6,7,8,9,10,11,12,13].

Regarding morphological aspects of the secondary gait disorder, consensus among researchers was reached for six gait patterns such as “drop foot”, “true equinus”, “apparent equinus” “genu recurvatum”, “jump gait”, and “crouch gait” considering only sagittal plane kinematics and disregarding functional ability [14]. Apart from that, the Gross Motor Function Classification System (GMFCS) is used to classify the degree of functional impairment, but provides no information on the underlying movement pathology [14,15]. Nevertheless, the assigned GMFCS level quite consistently stays stable in the long term [16].

In particular, unilateral CP is much more uncommon than bilateral CP and affected individuals are, on average, often less functionally impaired [17,18,19,20]. However, due to the natural asymmetry of unilateral CP, there are movement patterns that are highly complex, which are difficult to classify, and treatment recommendations can be difficult to provide [21,22].

Kinematic characteristics (including pelvic and trunk movements) as well as joint moments between the levels of functional impairment (represented by the specific GMFCS levels) may differ significantly, correlate with and even be able to contribute to the degree of the functional impairment and might therefore have an impact on decision making and treatment recommendations. Hence, the assumption arises that patients functioning in GMFCS level II show specific deviations and specific characteristics concerning kinematics and joint moments (e.g., more proximal involvement) that explain why these patients are more involved functionally than compared to GMFCS level I. Accordingly, specific therapy algorithms could be derived, if predominant and GMFCS level-specific movement pathologies can be identified. Ultimately, improving function, avoiding deterioration and improving quality of life are the main goals of orthopedic treatment [23].

To our knowledge, there is a lack of investigations analyzing GMFCS-specific kinematics and joint moments in all three planes, possibly missing relevant gait deviations. Therefore, the objective of this current work was to carry out a detailed analysis of kinematic features and joint moment analysis of individuals with unilateral CP (impaired side), taking all planes and dimensions of freedom of all lower limb joints, and additionally pelvic as well as trunk kinematics into account. Therefore, the aim is to identify GMFCS level-specific differences and the relationship between quantitative deviation and functional impairment in order to derive GMFCS level-specific treatment recommendations.

## 2. Materials and Methods

This study was conducted as a data base study following approval by the local ethics committee (S-198/2019).

Patients meeting the following *inclusion criteria* were included:Patients exclusively with unilateral CP;Functioning in GMFCS level I–II;No previous surgery of the lower limbs;No Botulinumtoxin–A injections within the last six months.

### 2.1. Study Population

In total, 89 individuals (40 female, 49 male) matched the inclusion criteria with a mean age at the time of instrumented 3D gait analysis (IGA) of 15.3 ± 9.6 years. All participants were classified according to their functional impairment using the GMFCS classification system [15]. A total of 63 patients were classified as GMFCS level I (patients with GMFCS level I showed the following distribution of gait patterns according to the classification system of Winters et al. [24]: 26 (41%) patients with type 1, 14 (22%) patients with type 2, 1 (2%) patient with type 3, 10 (16%) patients with type 4 and 12 (19%) patients were unclassified), whereas 26 patients met classification criteria according to GMFCS level II [6 (23%) patients with type 1, 5 (19%) patients with type 2, 1 (4%) patient with type 3, 11 (42%) patients with type 4 and 3 (12%) patients were unclassified).

### 2.2. Gait Analysis

Of the identified patients was performed from 2006 to 2017. For this a 120-Hz 9-camera system (Vicon, Oxford Metrics, Oxford, UK) and two piezoelectric force plates (Kistler, Winterthur, Switzerland) read out by a sampling frequency of 1080 Hz were used. Reflective markers were applied to bony landmarks according to the Plug-In Gait lower body model and protocol [25,26]. In this procedure, the knee axis was determined by the examiner via a knee alignment device. Four additional markers on the subjects’ shoulder girdle (processus spinosus of the 7th cervical vertebra, left and right acromion and incisura jugularis) were used to observe trunk motion in relation to the global reference frame [27]. All participants were asked to walk a seven-meter walkway barefoot and at a self-selected speed. The data were extracted from our motion laboratory data base.

### 2.3. Data Analysis

Motion data were processed via commercial software by Vicon using the Plug-In Gait model. For visual inspection of stride-to-stride consistency as well as time normalization of gait data to the gait cycle (GC in %), lab-specific software codes were used on the basis of Matlab R2018b (MathWorks, Natick, MA, USA). The following features considering all lower limb joints as well as the pelvis and trunk and all degrees of freedom have been included for further analysis:Kinematic parameters:○Ankle flexion/extension, ankle valgus/varus and foot progression;○Knee flexion/extension, knee valgus/varus and knee rotation;○Hip flexion/extension, hip abduction/adduction and hip rotation;○Pelvic tilt, pelvic obliquity and pelvic rotation;○Trunk tilt, trunk obliquity and trunk rotation.
Joint moments (internal joint moments are reported):○Ankle flexion/extension, valgus/varus and ankle rotation moments;○Knee flexion/extension, knee valgus/varus and knee rotation moments;○Hip flexion/extension, hip abduction/adduction and hip rotation moments.


Foot progression, describing the orientation of the foot’s long axis in relation to the gait direction, has been chosen instead of ankle rotation, since this parameter is given more clinical importance. The collected measures were compared against each other to investigate for potential differences between GMFCS levels, in order to assess for GMFCS level-specific characteristics and deviations from the gait of typically developing (TD) individuals. The reference data were derived from a group of TD, from our gait laboratory data base. The TD reference group consisted of 26 participants (56 limbs) with a mean age of 15.1 ± 5.9 years. The named sub-phases of gait were defined as follows: early stance phase (early StP) 0–10%, mid stance (MSt) 11–30%, late stance phase/preswing (late StP): 31–60%, 61–73% early swing phase (early SwP), mid swing (MSw) 74–87% and late swing phase (late SwP) 88–100%.

### 2.4. Statistical Analysis

Data were structured using Microsoft Excel (Microsoft, Redmond, WA, USA) and analyzed using Matlab R2018b (MathWorks, Natick, MA, USA). The mean and the standard deviation (SD) were calculated and displayed graphically. For comparative continuous statistics concerning kinematic features as well as joint moments, one-dimensional statistical parametric mapping (SPM) was performed with ANOVA-1D throughout the whole gait cycle using custom scripts in Matlab based on previous own work and on the work of Pataky et al. [28,29,30,31,32]. The level of significance was set at *p* < 0.05.

## 3. Results

Figure 1 and Figure 2 show the kinematics and joint moments across the gait cycle. Periods of significant differences between groups according to the SPM analysis are represented by black bars below the corresponding gait graph.

### 3.1. Kinematic Measures

There were no significant differences found between GMFCS levels regarding ankle kinematics for any plane. Both GMFCS levels I and II, showed equinus deformity at initial contact (IC) and a reduced dorsiflexion during StP and SwP with significant differences compared to the TD (Figure 1e1). In the coronal plane, levels I and II showed significant valgus during StP when compared to the TD (Figure 1e2). Foot progression angle values were shown to differ significantly from the TD, only for level I at IC and during MSt (Figure 1e3).

Knee flexion was pronounced in level I and II individuals at IC and loading response compared to the TD (Figure 1d1). Level II additionally showed significantly pronounced knee flexion during MSt (Figure 1d1). During SwP, range of motion was significantly reduced in GMFCS level I and II (Figure 1d1). Significant differences between level I and II were found during early SwP, and a reduced peak knee flexion was found at level II (Figure 1d1). In the coronal plane, the obtained knee kinematic values were significantly altered compared to the TD during the whole StP and during late SwP (Figure 1d2). No differences between GMFCS levels were found. Regarding knee rotation, values for GMFCS level I were mostly within the range of the TD, whereas GMFCS level II showed external rotation throughout almost the whole gait cycle (Figure 1d3). Here, differences between GMFCS level I and II were significant during early and late StP (Figure 1d3).

Hip flexion was significantly increased from MSt to early SwP for both GMFCS levels I and II (Figure 1c1). Statistically significant differences between level I and II were apparent for the StP, without reaching full hip extension at any time in level II (Figure 1c1). Hip abduction was reduced for GMFCS level I and II at late StP/early SwP compared to the TD, with no relevant differences between level I and II (Figure 1c2). In the transversal plane, there were no differences found between the different groups (Figure 1c3).

Both GMFCS levels I and II showed excessive anterior pelvic tilt from MSt to mid MSw compared to the TD without further differences between levels I and II (Figure 1b1). In the coronal plane, the obtained values showed no differences between level I and II with respect to pelvic obliquity, whereas both level I and II showed pelvic obliquity during early StP and additionally during SwP for level I (Figure 1b2). An excessive pelvic retraction was evident for both GMFCS level I and II during the whole gait cycle (Figure 1b3). There were no statistical differences between GMFCS level I and II in all three planes (Figure 1b1–b3).

Excessive anterior trunk tilt was evident for GMFCS level I and II compared to the TD during the majority of the gait cycle without significant differences between level I and II (Figure 1a1). Trunk obliquity was found to be ipsilaterally pronounced for GMFCS level II during StP, whereas level I was mainly within the range of the TD with significant differences between level I and II during StP (Figure 1a2). For level I, external trunk rotation mainly during SwP (corresponding to pelvic retraction) without significant differences compared to level II was evident (Figure 1a3).

### 3.2. Joint Moments

At ankle/foot level, both GMFCS levels I and II showed increased internal dorsiflexion moment during early StP and relatively increased plantarflexion moment/reduced dorsiflexion moment during MSt/late StP compared to the TD (Figure 2c1). Differences between level I and II were evident concerning peak dorsiflexion moments (Figure 2c1). In the coronal plane, joint moments for GMFCS level I and II were mainly within the range of the TD with increased varus moments during MSt and no differences between level I and II (Figure 2c2). With respect to transversally-acting moments, reduced internal rotation moments (Figure 2c3) with significantly reduced moments during StP for GMFCS levels I and II, compared to the TD, were found (Figure 2c3).

The obtained results for sagittal knee moments revealed reduced flexion as well as extension moments compared to the TD (Figure 2b1). Here, GMFCS level II showed significantly increased flexion moments during late StP compared to level I mostly without relevant extension moments (Figure 2b1). In the coronal plane, both level I and II showed reduced varus moments during early and late StP compared to the TD (Figure 2b2). Regarding knee rotation, reduced internal rotation moments were apparent for GMFCS level I and level II during StP (Figure 2b3). Here, internal rotation moments were significantly lower in GMFCS level II compared to level I (Figure 2b3).

At hip level, reduced extension moments during the majority of the gait cycle were especially evident for level I and II, compared to the TD, with no significant differences between level I and II (Figure 2a1). Hip adduction moments were reduced for level I and II during the majority of the gait cycle compared to the TD, without any relevant differences between level I and II (Figure 2a2). Regarding the transversal plane, there were reduced external rotation moments during early StP and reduced internal rotation moments during late StP for both level I and II, compared to the TD (Figure 2a3). In particular, hip external rotation moments were significantly increased for GMFCS level II compared to level I during late StP (Figure 2a3).

## 4. Discussion

Cerebral palsy represents a complex and continuous neurologic disorder with various phenotypes and degrees of severity. Other than bilateral CP, the natural asymmetry unilateral CP is characterized by a pronounced complexity of the underlying movement disorder. Although most patients with unilateral CP show less functional impairment than bilaterally affected patients, a relevant percentage show significant functional impairment.

The objective of this work was to assess kinematics as well as joint moment (as these are rarely analyzed [33]) parameters of patients exclusively with unilateral CP for potentially characteristic differences between GMFCS levels, in order to identify GMFCS level-specific deficits and specific treatment recommendations.

In summary, the results revealed that with both GMFCS levels I and II, individuals showed equinus deformity with especially reduced dorsiflexion moments, borderline ankle valgus and reduced internal ankle/foot rotation moments. At knee level, flexion with reduced range and reduced moments as well as knee valgus with reduced varus moments were evident. The obtained hip measures showed reduced extension moments with pronounced flexion, reduced internal rotation moments and reduced hip abduction. Further, anterior pelvic tilt, pelvic obliquity and pelvic retraction on the affected side were found. Compared to GMFCS level I, GMFCS level II was characterized by an additional pronounced reduction in dorsiflexion moments and pronounced external foot rotation moments at ankle level. Further, level II was additionally characterized by pronounced knee flexion with reduced range and a lack of relevant knee extension moments (weak extensors) and external knee rotation. At hip level, main differences concerned a pronounced flexion without extension during late StP (weak extensors), reduced hip abduction during late StP/early SwP and increased external rotation moments. The most striking and characteristic difference in GMFCS level II compared to level I was an additional excessive trunk obliquity/lean during StP.

Various authors have described equinus, in-toeing, stiff knee, excessive hip flexion and crouch to be the most prevalent gait abnormalities in hemiplegic patients [2] with excessive knee flexion and stiff knee deformity, hip internal rotation, excessive hip flexion and hip adduction as well as pronounced anterior pelvic tilt showing increasing prevalence between GMFCS level I and level II [19]. Our results compare to these findings. In our work, all patients analyzed by GMFCS levels showed various degrees of equinus, reduced knee range of motion and excessive knee and hip flexion. However, as our analysis included pelvic and trunk kinematics, pelvic retraction rather than in-toeing was found to be present in hemiplegic patients in GMFCS level I and II (Figure 1). Furthermore, a shift in power generation and peak moments from the ankle to the hip in hemiplegic individuals has been described by Riad et al. and Ishihara et al. and interpreted as a compensation mechanism for propulsion of walking [17,34]. Given the fact that our results suggest reduced extensor moments (ankle, knee and hip) and pronounced hip flexion moments our findings compare to those of the cited studies. In addition, reduced knee and hip joint extension were found to be associated with gait inefficiency in adolescents with CP [35]. Hence, therapeutic approaches improving joint extension are of crucial importance in decision making.

Additionally, rotational malalignment was found to be more common in GMFCS level II [19]. In contrast, our results suggest that pelvic retraction is equally present in GMFCS level I and II. Nevertheless, not only is pelvic retraction secondary to dynamic and anatomical internal hip rotation, often found in patients with unilateral CP, but it is additionally related to decreased ankle dorsiflexion, increased anterior pelvic tilt, Winters classification type II and asymmetrical posturing of the upper extremity during gait [36,37]. Here, (primary) internal hip rotation is hypothesized to also be compensatory in order to restore hip abductor moments [38,39,40]. However, internal hip rotation seems to not fully compensate for weak hip abductors, as often other compensatory movements (e.g., trunk lean) occur [38,41]. Furthermore, other additional deviations such as flexed knee gait (often associated with internal hip rotation and increasing the moments of the internal hip rotators) have to be taken into consideration [38,42].

Unilateral CP is thought to predominantly show distal involvement [43]. Our results suggest an (at least) additional proximal involvement, which is specific for individuals with GMFCS level II. The weakness of proximal muscles (hip extensors/abductors) especially leads to compensatory pelvic and trunk movements (Duchenne limp), which promote postural instability, explaining the impaired function particularly during stair climbing (GMFCS level I: walking indoors/outdoors, including running and jumping; climbing stairs without support; GMFCS level II: walking indoors/outdoors; climbing stairs with a railing). Other than in bilateral CP [44], in unilateral CP, excessive trunk lean seems to be apparent exclusively in patients with GMFCS level II. Comparable to our findings, two different hemiplegic pelvic/trunk motion patterns were identified, acting as different compensatory mechanisms for postural control deficits; a pro-gravitational gait pattern over-loading and an anti-gravitational gait under-loading the affected hip/limb [20,45,46]. In a subsequent study, the authors concluded, that other than in bilateral CP (postural/gait disorders in sagittal plane) patients with unilateral CP predominantly show postural/gait of disorders in coronal plane [47]. Even reduced ability to maintain balance due to somatosensory deficits plays a role here [48]. In general, Duchenne and Trendelenburg limping are thought to be a compensatory mechanism for unloading the hip abductors in patients with CP [27,41,49,50]. Excessive lateral trunk lean might be an effective way to compensate for weak hip abductors, though leading to drastically increased muscle effort and work and energy consumption [27,41]. However, at least for patients with bilateral CP, as trunk control is known to be impaired, trunk lean might not be considered as compensatory due to lower limb impairments, but also due to the underlying trunk control deficits [49,51]. The amount of trunk movement is linked with the severity of impairment, while additionally bilateral CP shows greater amounts for trunk movement [52]. From a clinician´s point of view, taking this into consideration, not only patients with unilateral CP and GMFCS level I (by restoring pelvic symmetry) but especially patients with unilateral CP and GMFCS level II, would benefit from a femoral derotation osteotomy (FDO) by addressing internal hip rotation/pelvic retraction and restoring hip abductor moments. Supracondylar FDO has been shown to increase frontal hip moments as changes in femoral anteversion directly influence hip kinetics and restore pelvic symmetry [38,42,53].

The main limitation of this current work is the absence of a differentiated multi-segment foot model for further exploration of the various foot deformities. In this study, individuals were classified according to the GMFCS classification system based on functional impairment. As a result, different gait patterns are included in the study groups of GMFCS level I and II. Furthermore, methodological differences among laboratories could lead to different gait measures compromising inter-study comparability. Nevertheless, IGA represents the most objective and sophisticated method for the evaluation of motion and gait disorders and is highly standardized.

## 5. Conclusions

In conclusion, in unilateral CP, pronounced reduction in extension moments and internal rotation moments at all joint levels as well as excessive trunk lean seem to be apparent and distinguishing features of patients with GMFCS level II compared to individuals with GMFCS level I. These patients would benefit particularly from FDO, not only by restoring pelvic symmetry such as in GMFCS level I patients, but also by improving hip abductor leverage. Further studies exploring potential GMFCS level-specific compensation mechanisms for the sound limb and GMFCS level-specific malalignment (static vs. dynamic, femoral vs. tibial) are necessary. With respect to classifying (unilateral) CP, inclusion of further segments, planes as well as pelvic and trunk kinematics might be beneficial for clinical decision making. Including multi-segment foot models and surface EMG as standard in future studies would be beneficial for identification of further subgroups, which may need modified treatment approaches [54,55]. Additionally, future studies should consider upper limb movement patterns and include upper limb measures, as there is a link to lower limb gait patterns and significance concerning postural stability, reduction in energy expenditure during gait and therapeutic value [56,57,58].

## Figures and Tables

**Figure 1 jcm-11-02556-f001:**
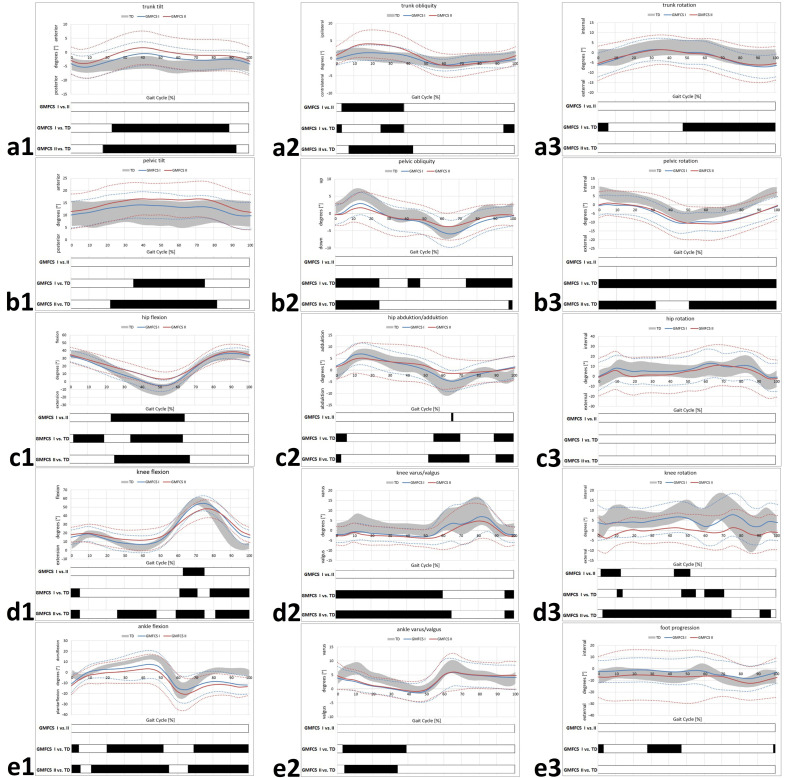
GMFCS level-specific kinematics. Trunk kinematics (**a1**–**a3**); pelvic kinematics (**b1**–**b3**); hip kinematics (**c1**–**c3**); knee kinematics (**d1**–**d3**); ankle/foot kinematics (**e1**–**e3**). **TD** group (age-matched typically developing individuals) of the gait laboratory data base. Black bars represent the results for **SPM** (statistical parametric mapping) and indicate periods of significant differences throughout of the gait cycle. **Straight lines** represent the mean of the involved limb of the participants with cerebral palsy (CP). **Dotted lines** indicate a range of ± 1 standard deviation (SD).

**Figure 2 jcm-11-02556-f002:**
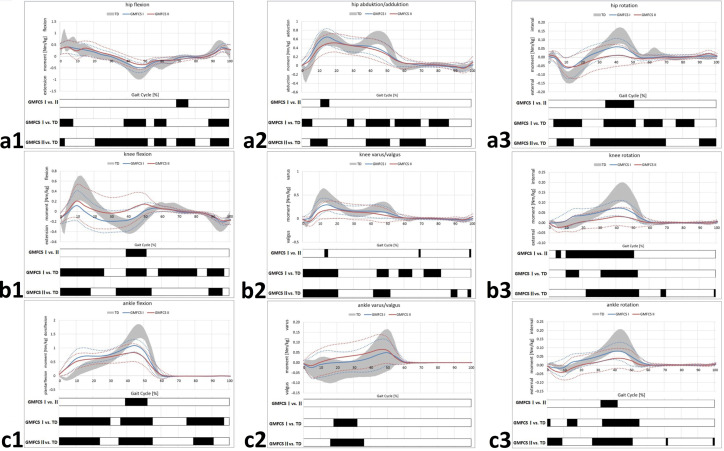
GMFCS level-specific joint moments. Hip moments (**a1**–**a3**); knee moments (**b1**–**b3**); ankle moments (**c1**–**c3**). **TD** group (age-matched typically developing individuals) of the gait laboratory data base. Black bars represent the results for **SPM** and indicate periods of significant differences throughout the gait cycle. **Straight lines** represent the mean of the involved limb of the participants with CP. **Dotted lines** indicate a range of ± 1 SD.

## Data Availability

Not applicable.

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
