# Peer review of "GMFCS Level-Specific Differences in Kinematics and Joint Moments of the Involved Side in Unilateral Cerebral Palsy"

_jcm, 2022, doi:10.3390/jcm11092556_

Round 1

Reviewer 1 Report

The authors investigated the kinematic and kinetic difference between children with unilateral CP with GMFCS levels. I have several comments that need to be addressed before the manuscript can be considered for publication.

Below I provide my thoughts on your manuscript.

Introduction

The rationale for assessing differences in kinematics between GMFCS levels was not clear from the introduction. What will the authors do with the information once differences have been found?

Additionally, it is of course expected from previous research that differences in kinematics will be found (based on gait type for instance [1]). It is also unclear which hypotheses the authors have based on this prior knowledge.

Furthermore, the authors have insufficiently described what is already know about kine(ma)tic differences between GMFCS levels of unilateral CP children (see for instance [2-5]

[1] Rodda J, Graham HK. Classification of gait patterns in spastic hemiplegia and spastic diplegia: a basis for a management algorithm. Eur J Neurol. [Case Reports Review]. 2001 Nov;8 Suppl 5:98-108

[2] Õunpuu S, Gorton G, Bagley A, Sison-Williamson M, Hassani S, Johnson B, Oeffinger D. Variation in kinematic and spatiotemporal gait parameters by Gross Motor Function Classification System level in children and adolescents with cerebral palsy. Dev Med Child Neurol. 2015 Oct;57(10):955-62. doi: 10.1111/dmcn.12766. Epub 2015 Apr 28. PMID: 25926016.

[3] Szopa A, Domagalska-Szopa M, Czamara A. Gait pattern differences in children with unilateral cerebral palsy. Res Dev Disabil. 2014 Oct;35(10):2261-6. doi: 10.1016/j.ridd.2014.05.020. Epub 2014 Jun 17. PMID: 24946266.

[4] Galli M, Cimolin V, Rigoldi C, Tenore N, Albertini G. Gait patterns in hemiplegic children with Cerebral Palsy: Comparison of right and left hemiplegia. Research in Developmental Disabilities, Volume 31, Issue 6, 2010, Pages 1340-1345, ISSN 0891-4222, doi.org/10.1016/j.ridd.2010.07.007.

[5] Agostini V, Nascimbeni A, Gaffuri A, Knaflitz M, Multiple gait patterns within the same Winters class in children with hemiplegic cerebral palsy, Clinical Biomechanics, Volume 30, Issue 9, 2015, Pages 908-914, ISSN 0268-0033, doi.org/10.1016/j.clinbiomech.2015.07.010.

--> i.e. What will this paper add to the existing literature?

Additionally, It is unclear why kinetic measures are included suddenly. They have not been introduced. The kinematics have been indirectly introduced via the different gait types but these are (up till now) only based on kinematics. What will the added value of the kinetics be in this study. And similarly to the previous question – what is new, but also what are the hypotheses?

The authors indicated in the introduction that compensations of the unimpaired side can occur, but in the final paragraph of the introduction they suddenly indicate that they will only focus on the impaired side. This is contradictory, and thus we will not gain insights in the possible more new aspect of compensations of the unaffected side.

The authors indicate in the final paragraph that they will focus on the kine(ma)tics of the lower limbs, pelvis and trunk – but they do not mention the upper limbs. Nevertheless it is known that the upper limb in unilateral CP is often more affected than the lower limb and also shows clear deviations during gait (see for instance [6-10]). Additionally there is a clear link between the upper and lower limb movements during gait (for more information see for instance a review which also has information on CP [11])

[6] Quantification of upper body strategy during gait in children with spastic diplegia using a summary parameter. Cimolin V, Condoluci C, Manzia CM, Girolamo GD, Galli M. Comput Methods Biomech Biomed Engin. 2020 Nov;23(15):1260-1266. doi: 10.1080/10255842.2020.1795144.

[7] A descriptive analysis of the upper limb patterns during gait in individuals with cerebral palsy.

Bonnefoy-Mazure A, Sagawa Y Jr, Lascombes P, De Coulon G, Armand S. Res Dev Disabil. 2014 Nov;35(11):2756-65. doi: 10.1016/j.ridd.2014.07.013.

[8] Altered arm posture in children with cerebral palsy is related to instability during walking.

Meyns P, Desloovere K, Van Gestel L, Massaad F, Smits-Engelsman B, Duysens J.

Eur J Paediatr Neurol. 2012 Sep;16(5):528-35. doi: 10.1016/j.ejpn.2012.01.011.

[9] Arm swing during walking at different speeds in children with Cerebral Palsy and typically developing children. Meyns P, Van Gestel L, Massaad F, Desloovere K, Molenaers G, Duysens J.

Res Dev Disabil. 2011 Sep-Oct;32(5):1957-64. doi: 10.1016/j.ridd.2011.03.029.

[10] Arm posture score and arm movement during walking: a comprehensive assessment in spastic hemiplegic cerebral palsy. Riad J, Coleman S, Lundh D, Broström E. Gait Posture. 2011 Jan;33(1):48-53. doi: 10.1016/j.gaitpost.2010.09.022.

[11] The how and why of arm swing during human walking. Meyns P, Bruijn SM, Duysens J. Gait Posture. 2013 Sep;38(4):555-62. doi: 10.1016/j.gaitpost.2013.02.006.

Methods ;

The two groups are not equal (balanced) which may have an effect on the spread of data within each group and thus affect possible statistical differences

The GMFCS groups were compared to a “TD referenc group consisted of 11 participants with a mean age of 23.0 years.”. First of all, there is a typo (should be reference). Second, the children are compared to an adult group which is thus not age-matched – which is not correct. Is this group matched for height and walking speed? If not, you are comparing apples and oranges (which you are actually already doing). Third, this group is even smaller – thus not balanced, which may affect possible statistical differences.

Statistics; a major issue is the apparent random choice of outcomes – these have not been justified according to the literature. Additionally, why did choose for several (random) gait events to statistically test for differences. To date, there are several statistical procedures to compare between different shapes of graphs (e.g. SPM). Considering the vast amount of outcomes that were selected, there are bound to be several statistical differences – this could be seen as data dredging (or data fishing). This combined with only showing only “a selection” of the findings in the tables/figures is worrying and indicates the lack of an actual clear research question and hypothesis. At the very least some kind of multiple testing correction should have been used.

Results

Unclear why only a selection was shown and not all the data. Also it is not described why this specific selection.

Discussion

The authors indicate that CP has various phenotypes and degrees of severity but in the actual investigation they only divide the group in 2 subgroups based on GMFCS. This is contradictory.

The authors state “The objective of this work was to assess kinematics as well as joint moments (as these are rarely analyzed [28])” – the fact that something is rarely analyzed is not a valid rationale to investigate it. It could be very well the case that this specific set of outcomes is irrelevant.

The authors state that they “identify GMFCS level-specific deficits and specific treatment recommendations.” It is however unclear how the findings of differences in kine(ma)tics between GMFCS level group of children with CP will result in specific treatment recommendations

I question what is actually a novel finding from the current study when reading the discussion.

Author Response

Author´s Reply to the Review Report (jcm-1616553)

Dear Dr. Emmanuel Andrès, dear Prof. Dr. Peter Choong, dear Reviewer,

first of all we want to thank the reviewer for the helpful comments regarding our submission through which we could improve the quality of our manuscript significantly.

The following list includes a detailed overview concerning the comments and the changes we made. The changes are highlighted in yellow color within the manuscript.

Answers to Reviewer 1:

We thank Reviewer 1 for the effort made and appreciate the constructive comments regarding our submission.

We would like to answer the suggestions and comments as follows:

Concerning the Introduction:

  1. The rationale for assessing differences in kinematics between GMFCS levels was not clear from the introduction. What will the authors do with the information once differences have been found?

Answer: Indeed, the rationale of assessing differences between GMFCS levels has not been clarified enough in the introduction. Therefore, we adjusted the last paragraph of the introduction specifying that deriving impacts on treatment recommendations (GMFCS level-specific) was the intention of our work.

  1. Additionally, it is of course expected from previous research that differences in kinematics will be found (based on gait type for instance [1]). It is also unclear which hypotheses the authors have based on this prior knowledge.

[1] Rodda J, Graham HK. Classification of gait patterns in spastic hemiplegia and spastic diplegia: a basis for a management algorithm. Eur J Neurol. [Case Reports Review]. 2001 Nov;8 Suppl 5:98-108

Answer: You are right. It is clear that differences e.g. between different gait types will be found. However, Rodda et al. for instance suggested subgroup-specific treatment regimens based on the classification system of Winters et al. (Winters, T.F., Jr.; Gage, J.R.; Hicks, R. Gait patterns in spastic hemiplegia in children and young adults. J. Bone Joint Surg. Am. 1987, 69, 437–441). This classification system is widely used and was developed for patients with spastic hemiplegia, examining 46 patients mainly—but not exclusively—with Cerebral Palsy (Winters, T.F., Jr.; Gage, J.R.; Hicks, R. Gait patterns in spastic hemiplegia in children and young adults. J. Bone Joint Surg. Am. 1987, 69, 437–441 AND Riad, J.; Haglund-Akerlind, Y.; Miller, F. Classification of spastic hemiplegic cerebral palsy in children. J. Pediatr. Orthop. 2007, 27, 758–764 AND McDowell, B.C.; Kerr, C.; Kelly, C.; Salazar, J.; Cosgrove, A. The validity of an existing gait classification system when applied to a representative population of children with hemiplegia. Gait Posture 2008, 28, 442–447.). Furthermore, established classification systems and previous research in this field mainly consider sagittal plane kinematics, disregarding transversal and coronal deviations, which are highly relevant and recently gained importance in this research field. The purpose of our study was to identify differences between different levels of functional impairment (depicted by GMFCS) that would correlate and possibly explain why e.g. patients functioning in GMFCS level II are more involved functionally than others. To our knowledge, this has not been investigated (including kinematics and kinetics in all three planes) sufficiently so far. In our work, exclusively patients with GMFCS level II showed trunk lean/Duchenne limping, which is able to explain why these patients function significantly worse than patients with GMFCS level I.

We added a hypothesis to our introduction to specify the purpose of our work.

  1. Furthermore, the authors have insufficiently described what is already know about kine(ma)tic differences between GMFCS levels of unilateral CP children (see for instance [2-5]

[2] Õunpuu S, Gorton G, Bagley A, Sison-Williamson M, Hassani S, Johnson B, Oeffinger D. Variation in kinematic and spatiotemporal gait parameters by Gross Motor Function Classification System level in children and adolescents with cerebral palsy. Dev Med Child Neurol. 2015 Oct;57(10):955-62. doi: 10.1111/dmcn.12766. Epub 2015 Apr 28. PMID: 25926016.

[3] Szopa A, Domagalska-Szopa M, Czamara A. Gait pattern differences in children with unilateral cerebral palsy. Res Dev Disabil. 2014 Oct;35(10):2261-6. doi: 10.1016/j.ridd.2014.05.020. Epub 2014 Jun 17. PMID: 24946266.

[4] Galli M, Cimolin V, Rigoldi C, Tenore N, Albertini G. Gait patterns in hemiplegic children with Cerebral Palsy: Comparison of right and left hemiplegia. Research in Developmental Disabilities, Volume 31, Issue 6, 2010, Pages 1340-1345, ISSN 0891-4222, doi.org/10.1016/j.ridd.2010.07.007.

[5] Agostini V, Nascimbeni A, Gaffuri A, Knaflitz M, Multiple gait patterns within the same Winters class in children with hemiplegic cerebral palsy, Clinical Biomechanics, Volume 30, Issue 9, 2015, Pages 908-914, ISSN 0268-0033, doi.org/10.1016/j.clinbiomech.2015.07.010.

--> i.e. What will this paper add to the existing literature?

Answer: Thank you for your constructive suggestions. Õunpuu et al. exclusively analyzed patients with bilateral CP. Szopa et al. mainly analyzed subgroup-specific GGI-values and paedobarographic measurements, not showing kinematic outcomes specifically. However, the findings of Szopa et al. have been discussed in the Discussion, as they identified one movement pattern under- and one over-loading the affected side in unilateral CP corresponding to the excessive trunk lean seen in our work. Galli et al. assessed differences between left and right hemiplegia regardless the GMFCS level. Furthermore, there have been inconsistent reports on the relevance of which side is affected in hemiplegic CP. Agostini et al. conducted a study analyzing foot–floor contact sequences and EMG-data reporting only sagittal ankle kinematics in patients with unilateral CP classified according to Winters et al., which therefore is not comparable to our methodology. None of the above mentioned studies included kinetic measures.

Please also see answer to comment no. 2.

To our knowledge, there is a lack of investigations including kinematics and joint moments in all three planes, possibly missing relevant gait deviations. This has been additionally stated in the Introduction.

  1. Additionally, It is unclear why kinetic measures are included suddenly. They have not been introduced. The kinematics have been indirectly introduced via the different gait types but these are (up till now) only based on kinematics. What will the added value of the kinetics be in this study. And similarly to the previous question – what is new, but also what are the hypotheses?

Answer: Thank you for your advice as a neutral reader. Kinetic measures and the intention to analyze those were first mentioned in the Introduction describing the objective of the current study. From our point of view, kinetic measures are relevant and important to analyze supplementary to kinematic measures, as they indicate e.g. hip abductor weakness and might have impact on treatment recommendations as stated in the Discussion (e.g. Thielen, M.; Wolf, S.I.; Klotz, M.C.M.; Geisbüsch, A.; Putz, C.; Krautwurst, B.; Dreher, T. Supracondylar femoral rotation osteotomy affects frontal hip kinetics in children with bilateral cerebral palsy. Dev Med Child Neurol 2019, 61, 322-328). We added a hypothesis to our introduction to specify the purpose of our work, which can be seen as kind of an exploratory data analysis.

  1. The authors indicated in the introduction that compensations of the unimpaired side can occur, but in the final paragraph of the introduction they suddenly indicate that they will only focus on the impaired side. This is contradictory, and thus we will not gain insights in the possible more new aspect of compensations of the unaffected side.

Answer: Thank you for your constructive suggestion. You are right. This paragraph was written to help the narrative and to outline major differences to bilateral Cerebral Palsy. Due to the naturally asymmetry of the disorder gait pathologies in unilateral Cerebral Palsy are more complex. However, compensatory mechanisms of the uninvolved limb are subject to current investigations.

  1. The authors indicate in the final paragraph that they will focus on the kine(ma)tics of the lower limbs, pelvis and trunk – but they do not mention the upper limbs. Nevertheless it is known that the upper limb in unilateral CP is often more affected than the lower limb and also shows clear deviations during gait (see for instance [6-10]). Additionally there is a clear link between the upper and lower limb movements during gait (for more information see for instance a review which also has information on CP [11])

[6] Quantification of upper body strategy during gait in children with spastic diplegia using a summary parameter. Cimolin V, Condoluci C, Manzia CM, Girolamo GD, Galli M. Comput Methods Biomech Biomed Engin. 2020 Nov;23(15):1260-1266. doi: 10.1080/10255842.2020.1795144.

[7] A descriptive analysis of the upper limb patterns during gait in individuals with cerebral palsy. Bonnefoy-Mazure A, Sagawa Y Jr, Lascombes P, De Coulon G, Armand S. Res Dev Disabil. 2014 Nov;35(11):2756-65. doi: 10.1016/j.ridd.2014.07.013.

[8] Altered arm posture in children with cerebral palsy is related to instability during walking. Meyns P, Desloovere K, Van Gestel L, Massaad F, Smits-Engelsman B, Duysens J. Eur J Paediatr Neurol. 2012 Sep;16(5):528-35. doi: 10.1016/j.ejpn.2012.01.011.

[9] Arm swing during walking at different speeds in children with Cerebral Palsy and typically developing children. Meyns P, Van Gestel L, Massaad F, Desloovere K, Molenaers G, Duysens J. Res Dev Disabil. 2011 Sep-Oct;32(5):1957-64. doi: 10.1016/j.ridd.2011.03.029.

[10] Arm posture score and arm movement during walking: a comprehensive assessment in spastic hemiplegic cerebral palsy. Riad J, Coleman S, Lundh D, Broström E. Gait Posture. 2011 Jan;33(1):48-53. doi: 10.1016/j.gaitpost.2010.09.022.

[11] The how and why of arm swing during human walking. Meyns P, Bruijn SM, Duysens J. Gait Posture. 2013 Sep;38(4):555-62. doi: 10.1016/j.gaitpost.2013.02.006.

Answer: You are right. In unilateral CP, patients often show more involvement at the upper limbs. Nevertheless, our intention in this work was to further investigate lower limb, pelvis and trunk features during gait in all three planes as usually mainly sagittal plane features of ankle, knee and hip joint are reported and analyzed as a standard in research articles in this field. We additionally decided to consider and focus on joint moments (see also answer to comment no. 12) as supplementary features instead of upper limb kinematics. However, we included upper limb movement patterns and their link to lower limb during gait in our Conclusion section as implications for future studies.

Concerning the Methods section:

  1. The two groups are not equal (balanced) which may have an effect on the spread of data within each group and thus affect possible statistical differences

Answer: Indeed, the two groups are not equal in numbers, which is due to and representative of the natural prevalence of the different impairment levels in unilateral Cerebral Palsy. However, our intention was to assess untreated patients with unilateral Cerebral Palsy, which are rare to encounter. As we stated in the Methods sections, instrumented gait analyses from 2006 – 2017 were considered, resulting in a total number of 89 patients. Adjusting/Balancing the two subgroups with respect to numbers would have led to a much smaller cohort size in total and would bias the spread of data of patients with GMFCS level I. This is why we chose to include all available gait analyses of untreated patients with unilateral Cerebral Palsy.

  1. The GMFCS groups were compared to a “TD referenc group consisted of 11 participants with a mean age of 23.0 years.”. First of all, there is a typo (should be reference). Second, the children are compared to an adult group which is thus not age-matched – which is not correct. Is this group matched for height and walking speed? If not, you are comparing apples and oranges (which you are actually already doing). Third, this group is even smaller – thus not balanced, which may affect possible statistical differences.

Answer: We would like to thank the Reviewer for this constructive and very helpful suggestion. You are right. As a result we created a new and age-matched reference group consisting of 28 participants (56 limbs) in order to improve statistics and thus the quality of our work. The Methods and Results section as well as the Discussion have been adapted accordingly.

  1. Statistics; a major issue is the apparent random choice of outcomes – these have not been justified according to the literature. Additionally, why did choose for several (random) gait events to statistically test for differences. To date, there are several statistical procedures to compare between different shapes of graphs (e.g. SPM). Considering the vast amount of outcomes that were selected, there are bound to be several statistical differences – this could be seen as data dredging (or data fishing). This combined with only showing only “a selection” of the findings in the tables/figures is worrying and indicates the lack of an actual clear research question and hypothesis. At the very least some kind of multiple testing correction should have been used.

Answer: We would like to thank the Reviewer for this constructive and suggestion. You are right. The selection of outcomes chosen was not justified/clarified enough in the manuscript. Our choice was not random, as we tried to go along with previous research of other groups and publications as an aspect of orientation to allow inter-study comparability (for instance minimum knee flexion angle during stance to assess the amount of “crouch gait”; Õunpuu S, Gorton G, Bagley A, Sison-Williamson M, Hassani S, Johnson B, Oeffinger D. Variation in kinematic and spatiotemporal gait parameters by Gross Motor Function Classification System level in children and adolescents with cerebral palsy. Dev Med Child Neurol. 2015 Oct;57(10):955-62. doi: 10.1111/dmcn.12766. Epub 2015 Apr 28. PMID: 25926016). However, you are right in saying that this was not justified/clarified. Furthermore, we chose not to submit the total of all outcomes for every gait event and gait phase, due to the excessive set of data. Instead, as you suggested, we performed SPM including the reference group to improve the statistics of our work. The expressed concerns regarding a “lack of a clear research question” was addressed by adding a hypothesis to our introduction.

Concerning the Results section:

  1. Unclear why only a selection was shown and not all the data. Also it is not described why this specific selection.

Answer: You are right. As stated in our answer concerning comment no. 9, our choice was not random, as we tried to go along with previous research of other groups and publications. However, this was not clarified. We chose not to submit the total of all outcomes for every gait event and gait phase as Appendices, due to the excessive set of data that partially showed no statistically significant results. However, SPM was performed instead in order to improve the statistics and the Results section. Also, please see our answer to comment no. 9.

Concerning the Discussion:

  1. The authors indicate that CP has various phenotypes and degrees of severity but in the actual investigation they only divide the group in 2 subgroups based on GMFCS. This is contradictory.

Answer: You are right. The first paragraph of the Discussion section was chosen like this to help the narrative. However, in the Introduction as well as in the Discussion we reworded according paragraphs focusing on “degrees of severity” only.

  1. The authors state “The objective of this work was to assess kinematics as well as joint moments (as these are rarely analyzed [28])” – the fact that something is rarely analyzed is not a valid rationale to investigate it. It could be very well the case that this specific set of outcomes is irrelevant.

Answer: You are right. The fact that something is rarely analyzed does not mean that is relevant. However, disregarding a set of outcomes could miss relevant differences/characteristics. For instance, current classifications systems concerning CP mainly focus on sagittal plane kinematics and consensus among researchers was reached only for sagittal plane features (Papageorgiou, E.; Nieuwenhuys, A.; Vandekerckhove, I.; Van Campenhout, A.; Ortibus, E.; Desloovere, K. Systematic review on gait classifications in children with cerebral palsy: An update. Gait Posture 2019, 69, 209–223.). In the past it was assumed that e.g. the so-called “0-group” in unilateral CP showed irrelevant/scarcely detectable gait pathologies (Riad, J.; Haglund-Akerlind, Y.; Miller, F. Classification of spastic hemiplegic cerebral palsy in children. J. Pediatr. Orthop. 2007, 27, 758–764 AND McDowell, B.C.; Kerr, C.; Kelly, C.; Salazar, J.; Cosgrove, A. The validity of an existing gait classification system when applied to a representative population of children with hemiplegia. Gait Posture 2008, 28, 442–447.). However, recently it was shown that there are indeed changes in the previously neglected transversal plane for instance. Furthermore, it could be shown that rotational malalignment is on the one hand a consequence of spasticity and on the other hand an attempt to improve weak hip abductor levers (Thielen, M.; Wolf, S.I.; Klotz, M.C.M.; Geisbüsch, A.; Putz, C.; Krautwurst, B.; Dreher, T. Supracondylar femoral rotation osteotomy affects frontal hip kinetics in children with bilateral cerebral palsy. Dev Med Child Neurol 2019, 61, 322-328, doi:10.1111/dmcn.14035). In order not to miss indications of e.g. weak extensor muscles, we decided to analyze joint moments as a supplement to gait kinematics.

  1. The authors state that they “identify GMFCS level-specific deficits and specific treatment recommendations.” It is however unclear how the findings of differences in kine(ma)tics between GMFCS level group of children with CP will result in specific treatment recommendations.

Answer: As stated in the second last paragraph of the Discussion and as stated in the Conclusion most remarkable and characteristic feature of patients functioning in GMFCS level II was trunk lean during stance phase, which is seen as compensation for insufficiency of hip abductors. We concluded that femoral derotation osteotomy on the one hand would restore pelvic asymmetry and on the other hand could improve hip abductor leverage as shown previously (Thielen, M.; Wolf, S.I.; Klotz, M.C.M.; Geisbüsch, A.; Putz, C.; Krautwurst, B.; Dreher, T. Supracondylar femoral rotation osteotomy affects frontal hip kinetics in children with bilateral cerebral palsy. Dev Med Child Neurol 2019, 61, 322-328, doi:10.1111/dmcn.14035). As a result of the current work, own future investigations will consider this as a research hypothesis analyzing patients with unilateral CP that underwent femoral derotation osteotomy.

We hope that our manuscript now, after profound revision, accomplishes the high standards of Journal of Clinical Medicine.

We are looking forward to hearing from you.

Yours sincerely,

Stefanos Tsitlakidis                       and                          Marco Götze

(Submitting author)                                                      (Corresponding author)

Reviewer 2 Report

In the current study, the authors analyze differences in joint kinematics and kinetics between GMFCS I and II patients with unilateral CP. In my opinion, there are some major concerns that need to be addressed. With this concerns (please find my comments below), I think that conclusions cannot yet be drawn from the results.

First of all, you have two groups of patients with GMFCS level I and II. However, you did not distinguish between different gait patterns in these groups. You have mentioned the different gait patterns observed in CP patients as well as within unilateral CP patients (e.g. Wren et al, 2005). Therefore, I am wondering if you do not loose information by mixing patients with different gait patterns? E.g. the mean of knee hyperextension and increased knee flexion in terminal stance could appear like normal knee extension in this phase. The literature shows e.g. that for the evaluation of the efficacy of ankle-foot-orthosis in CP patients the consideration of gait patterns is essential. At least, it would be great to have a number of different gait patterns present in both groups. Just something to think about: wouldn’t it be possible to e.g. compare patients with one distinct gait pattern in GMFCS I and II against each other?

My second major concern is about data analysis. You can find several comments addressing this part below.

Detailed comments:

Is the sampling frequency for your force plates the same as for the camera system?

You state that markers were applied to bony landmarks according to the protocol of Kadaba et al. (1990). Actually, in this publication the foot is described as a vector “line segment joining the ankle center and the marker at the foot (between the second and third metatarsal heads)”. The Plug-In Gait model has an additional Heel-marker. My main point, however, is, if you feel certain that with the mentioned marker set frontal or transversal plane kinematics/kinetics of the ankle can be adequately addressed?

There is also information missing based on which algorithm the calculations of joint kinematics and kinetics were performed (hip and knee rotation based on wand markers?). This might also answer my previous question. In terms of completeness, you might also want to briefly mention data processing; e.g. data filtering, normalization of joint moment,… (or at least cite another study where it was performed in the same way). Please also mention if your joint moments presented are external or internal moments.

What does high/low foot progression angle mean? Is this the foot progression angle relative to the laboratory coordinate system? Thus, high being internal and low being external? If so, please consider not mentioning it together with the ankle joint kinematics to avoid confusion.

Have you considered using a statistical method, which allows for analysis of the whole gait cycle (e.g. statistical parametric mapping (SPM)) rather than pre-defined time points as well as maxima, minima and range of motion?

This goes along with my next question: how did you select the different phases/parameters analyzed? Following some examples: E.g. you define mid swing as one of the phases of interest. The only parameter analyzed in mid swing is the maximum knee flexion moment. Why is it necessary to look at the maximum in mid swing rather than over the total swing phase? Why are you not interested in e.g. knee flexion in terminal stance? You looked at the minimum knee flexion angle in stance phase, however, depending on the gait pattern this could occur either at initial contact or in terminal stance.

Why was knee rotation assessed at three different time points? Do you expect it to change over that gait cycle duration?

Statistics:

  • If you analyse the various parameters within one gait cycle, should you not correct the p-value for multiple comparison?
  • You are presenting your results in comparison to TD as well. Therefore, I suggest including TD in your statistical evaluation as well.
  • g. with the use of SPM you could compare all three groups and you no longer have to consider my first comment concerning statistics). I believe this manuscript could greatly benefit from such a method of analysis.
  • The presentation of effect sizes would be helpful for a better understand of clinical relevance. E.g. hip adduction at toe off would only have a small effect size (Cohens d) and might not be clinical relevant.

What is the functional meaning of rotational moments for you?

Figures:

Figures are very hard to read and interpret. Please consider making new graphs. Please find below some questions/comments concerning the graphs:

  • Knee adduction moment / y axis label is not correct, since you write “valgus” for positive and negative values
  • Presentation of reference data (which you might want to call TD to go along with the manuscript) would be great as shaded area for easier interpretation
  • Why presenting each 5% step of patient’s data with a dot? Does it have a meaning?

Discussion: since I have doubts concerning the analysis of the results (comments above), I am unsure whether all conclusion drawn can be made at this stage. At least, a more hypothetical interpretation might be necessary.

Just picking one example: talking about femoral derotation osteotomy – this conclusion is drawn from average gait data over various gait pattern. Wouldn’t it be more precise to compare GMFCS I and II patients that actually show a gait pattern where hip rotation/pelvic asymmetry plays a role to conclude if GMFCS II would especially benefit?

Author Response

Author´s Reply to the Review Report (jcm-1616553)

Dear Dr. Emmanuel Andrès, dear Prof. Dr. Peter Choong, dear Reviewer,

first of all we want to thank the reviewer for the helpful comments regarding our submission through which we could improve the quality of our manuscript significantly.

The following list includes a detailed overview concerning the comments and the changes we made. The changes are highlighted in yellow color within the manuscript.

Answers to Reviewer 2:

We thank Reviewer 1 for the effort made and appreciate the constructive comments regarding our submission.

We would like to answer the suggestions and comments as follows:

  1. First of all, you have two groups of patients with GMFCS level I and II. However, you did not distinguish between different gait patterns in these groups. You have mentioned the different gait patterns observed in CP patients as well as within unilateral CP patients (e.g. Wren et al, 2005). Therefore, I am wondering if you do not loose information by mixing patients with different gait patterns? E.g. the mean of knee hyperextension and increased knee flexion in terminal stance could appear like normal knee extension in this phase. The literature shows e.g. that for the evaluation of the efficacy of ankle-foot-orthosis in CP patients the consideration of gait patterns is essential. At least, it would be great to have a number of different gait patterns present in both groups. Just something to think about: wouldn’t it be possible to e.g. compare patients with one distinct gait pattern in GMFCS I and II against each other?

Answer: Thank you for your advice as a neutral reader. You are right. Analyzing patients according to their functional impairment (by GMFCS levels) disregards morphologic subtypes and gait patterns. In our paragraph concerning the limitations of our work, we outlined this limitation. However, an association between gait types and functional involvement was described in the past.[1-2] As a result, patients in GMFCS level I would mainly be classified as type 1 or type 2 (which are close to each other) and patients in GMFCS level II largely as type 4 according to Winters et al. Moreover, subdividing specific gait patterns by GMFCS levels would shrink sample sizes significantly in this in general small population (untreated patients with unilateral CP).[2] The purpose of this current study was to identify differences between different levels of functional impairment (depicted by GMFCS) that would correlate and explain why e.g. patients functioning in GMFCS level II are more involved functionally than others. To our knowledge, this has not been investigated so far for unilateral CP. Ounpuu et al. conduced a similar study exclusively including patients with bilateral CP (but only sagittal plane kinematics).[3] In our work, exclusively patients with GMFCS level II showed trunk lean/Duchenne limping, which is able to explain why these patients function significantly worse than patients with GMFCS level I.

[1] Winters, T.F., Jr.; Gage, J.R.; Hicks, R. Gait patterns in spastic hemiplegia in children and young adults. J. Bone Jt. Surg. Am. 1987, 69, 437–441

[2] Tsitlakidis, S.; Horsch, A.; Schaefer, F.; Westhauser, F.; Goetze, M.; Hagmann, S.; Klotz, M.C.M. Gait Classification in Unilateral Cerebral Palsy. J. Clin. Med. 2019, 8, 1652

[3] Õunpuu S, Gorton G, Bagley A, Sison-Williamson M, Hassani S, Johnson B, Oeffinger D. Variation in kinematic and spatiotemporal gait parameters by Gross Motor Function Classification System level in children and adolescents with cerebral palsy. Dev Med Child Neurol. 2015 Oct;57(10):955-62

  1. Is the sampling frequency for your force plates the same as for the camera system?

Answer: Sampling frequency is 1080 Hz i.e. 9 times the camera frequency.

  1. You state that markers were applied to bony landmarks according to the protocol of Kadaba et al. (1990). Actually, in this publication the foot is described as a vector “line segment joining the ankle center and the marker at the foot (between the second and third metatarsal heads)”. The Plug-In Gait model has an additional Heel-marker. My main point, however, is, if you feel certain that with the mentioned marker set frontal or transversal plane kinematics/kinetics of the ankle can be adequately addressed?

Answer: To our knowledge the protocol of Kadaba et al. and the Plug-In Gait model are nearly identical including the heel marker. Frontal ankle kinematics were added later. However, Plug-In Gait/Kadaba et al. still is the reference method. Only inclusion of a specific multi-segment foot-model would improve accuracy. The methods section has been adapted to specifically mention the use of the Plug-In Gait model.

  1. There is also information missing based on which algorithm the calculations of joint kinematics and kinetics were performed (hip and knee rotation based on wand markers?). This might also answer my previous question. In terms of completeness, you might also want to briefly mention data processing; e.g. data filtering, normalization of joint moment,… (or at least cite another study where it was performed in the same way). Please also mention if your joint moments presented are external or internal moments.

Answer: We did not perform filtering but averaging at least 5 strides. Normalization to body weight was realized. As reference citation please see [4]. The Methods section was adapted accordingly.

[4] Davis, R. B., et al. (1991). "A gait analysis data collection and reduction technique." Hum Mov Sci 10(5): 575-587

  1. What does high/low foot progression angle mean? Is this the foot progression angle relative to the laboratory coordinate system? Thus, high being internal and low being external? If so, please consider not mentioning it together with the ankle joint kinematics to avoid confusion.

Answer: “Foot progression” as long foot axis in relation to the gait direction. “High” or positive values being intoeing. Standard reporting next to ankle rotation.

  1. Have you considered using a statistical method, which allows for analysis of the whole gait cycle (e.g. statistical parametric mapping (SPM)) rather than pre-defined time points as well as maxima, minima and range of motion?

Answer: We would like to thank the Reviewer for this constructive and very helpful suggestion. According to your suggestion and according to those of the other reviewers, we have performed SPM and adjusted the Methods section, the Results section (including Figures and removing Tables 1 and 2) and the Discussion accordingly, to improve the statistics and quality of our work.

  1. This goes along with my next question: how did you select the different phases/parameters analyzed? Following some examples: E.g. you define mid swing as one of the phases of interest. The only parameter analyzed in mid swing is the maximum knee flexion moment. Why is it necessary to look at the maximum in mid swing rather than over the total swing phase? Why are you not interested in e.g. knee flexion in terminal stance? You looked at the minimum knee flexion angle in stance phase, however, depending on the gait pattern this could occur either at initial contact or in terminal stance.

Answer: We would like to thank the Reviewer for this constructive and very helpful suggestion. Our choice was not random, as we tried to go along with previous research and publications. Parameters, phases and gait events were chosen according to publications of other authors to allow/enhance inter-study comparability. (for instance minimum knee flexion angle during stance to assess the amount of “crouch gait”). However, the selection of outcomes chosen was not justified/clarified enough in the manuscript. In order to improve the statistics of our manuscript SPM was been performed according to your suggestion. Further, please see our answer to comment no. 6.

  1. Why was knee rotation assessed at three different time points? Do you expect it to change over that gait cycle duration?

Answer: Please see our answers to comments no. 6 and no. 7. As suggested we performed SPM to improve the statistics of manuscript.

  1. You are presenting your results in comparison to TD as well. Therefore, I suggest including TD in your statistical evaluation as well. With the use of SPM you could compare all three groups and you no longer have to consider my first comment concerning statistics). I believe this manuscript could greatly benefit from such a method of analysis.

Answer: We would like to thank the Reviewer for this constructive and very helpful suggestion. We created a bigger and better age-matched TD group that was, according to your suggestions, included in SPM (also see answer to comments no. 6 and 8). In order to improve the statistics of our work and to meet your suggestions, we replaced t-test results shown in Tables 1 and 2 with the SPM-results represented by black bars in the newly designed Figures 1 and 2.

  1. The presentation of effect sizes would be helpful for a better understand of clinical relevance. E.g. hip adduction at toe off would only have a small effect size (Cohens d) and might not be clinical relevant.

Answer: Please see our answers to comments no. 6 and no. 7. As suggested we performed SPM to improve the statistics of manuscript.

  1. What is the functional meaning of rotational moments for you?

Answer: Hip rotation and pelvic asymmetry play a significant role in almost all patients with unilateral CP regardless the gait type according to recent research and in house ongoing investigations (see also answer to comment no.14).[5]. Internal hip rotation (moments) have been described to be associated with increased plantarflexion moments.[6] Further, rotational malalignment (specifically increased femoral antetorsion that is compensated through internal hip rotation) lead to secondary abductor insufficiency.[7]. Both, increased plantarflexion moments and hip abductor insufficiency cause functional impairment and often are compensated on another joint or segment level (e.g. pelvic retraction)  Moreover, since usually current classification systems disregard cornal and transversal planes, rotational moments were assessed for reasons of completeness.

[5] Riad, J.; Finnbogason, T.; Broström, E. Anatomical and dynamic rotational alignment in spastic unilateral cerebral palsy. Gait Posture 2020, 81, 153-158, doi:10.1016/j.gaitpost.2020.07.010

[6] Brunner R, Taylor WR, Visscher RMS. Restoration of Heel-Toe Gait Patterns for the Prevention of Asymmetrical Hip Internal Rotation in Patients with Unilateral Spastic Cerebral Palsy. Children (Basel). 2021 Sep 2;8(9):773.

[7] Thielen, M.; Wolf, S.I.; Klotz, M.C.M.; Geisbüsch, A.; Putz, C.; Krautwurst, B.; Dreher, T. Supracondylar femoral rotation osteotomy affects frontal hip kinetics in children with bilateral cerebral palsy. Dev Med Child Neurol 2019, 61, 322-328, doi:10.1111/dmcn.14035

  1. Knee adduction moment / y axis label is not correct, since you write “valgus” for positive and negative values

Answer: We corrected the y axis label in the newly designed Figure 2.

  1. Presentation of reference data (which you might want to call TD to go along with the manuscript) would be great as shaded area for easier interpretation. Why presenting each 5% step of patient’s data with a dot? Does it have a meaning?

Answer: Thank you for your advice as a neutral reader and your suggestions. We adjusted the style of the graphs/figures to improve the presentation and interpretation by displaying the reference data as a shaded area and by removing the dots at every 5% (as they had no specific meaning).

  1. Since I have doubts concerning the analysis of the results (comments above), I am unsure whether all conclusion drawn can be made at this stage. At least, a more hypothetical interpretation might be necessary. Just picking one example: talking about femoral derotation osteotomy – this conclusion is drawn from average gait data over various gait pattern. Wouldn’t it be more precise to compare GMFCS I and II patients that actually show a gait pattern where hip rotation/pelvic asymmetry plays a role to conclude if GMFCS II would especially benefit?

Answer: We would like to thank the Reviewer for this constructive comment. As described in our answer to comment no. 1, patients in GMFCS level I are predominantly classified as type 1 or type 2 (which are close to each other) and patients in GMFCS level II as type 4 according to Winters et al. Subdividing specific gait patterns by GMFCS levels would shrink sample sizes significantly in this in general small population negatively affecting statistical analysis and interpretability.

Moreover, established classification systems for unilateral CP (especially the system of Winters et al. or Rodda et al.) and more general multiple joint patterns that have reached consensus among researchers [8] disregard coronal and transversal plane deviations. Recent research and in house ongoing investigations show/indicate that hip rotation and pelvic asymmetry play a significant role in almost all patients with unilateral CP regardless the gait type, even in the mildly involved patients, that initially were thought to be irrelevantly involved just showing subtile gait deviations and therefore were classified as the “0-group”.[9-11] However, as described in our answers concerning comments no. 6 and no. 7, we performed SPM including all three groups.

[8] Papageorgiou, E.; Nieuwenhuys, A.; Vandekerckhove, I.; Van Campenhout, A.; Ortibus, E.; Desloovere, K. Systematic review on gait classifications in children with cerebral palsy: An update. Gait Posture 2019, 69, 209–223)

[9] Riad, J.; Haglund-Akerlind, Y.; Miller, F. Classification of spastic hemiplegic cerebral palsy in children. J. Pediatr. Orthop. 2007, 27, 758–764

[10] Riad, J.; Finnbogason, T.; Broström, E. Anatomical and dynamic rotational alignment in spastic unilateral cerebral palsy. Gait Posture 2020, 81, 153-158

[11] Tsitlakidis S, Schwarze M, Westhauser F, Heubisch K, Horsch A, Hagmann S, Wolf SI, Götze M. Gait Indices for Characterization of Patients with Unilateral Cerebral Palsy. J Clin Med. 2020 Nov 30;9(12):3888

We hope that our manuscript now, after profound revision, accomplishes the high standards of Journal of Clinical Medicine.

We are looking forward to hearing from you.

Yours sincerely,

Stefanos Tsitlakidis                          and                    Marco Götze

(First author)                                                              (Corresponding author)

Reviewer 3 Report

This work describes gait features of children with unilateral CP comparing GMFCS I and II. The work is of interest for clinicians and researchers that work in this fields but there are several major issues to be addressed. Please find below my comments.

Major comments:

  • Authors analysed and reported data about the impaired side of patients included in the analysis. It would have been of interest to include in the report also the unaffected limb and its ability and limitations to execute compensation as suggested by authors in the Introduction section. If this not desired fore some reasons, I suggest to specify in the title and abstract that the paper describe only the impaired side.
  • The Materials and methods section has to be restructured: it contains part of results, references to figures and table and does not include important sections such as inclusion and exclusion criteria (that are reported I know but between brackets). Define inclusion criteria before presenting the sample size (that I would move in the results section) (lines 74-79). Consider to use sub-paragraphs to describe 1) the equipment; 2)the population 3) the data analysis (this part is missing and must be added. How did you extracted parameters? Did you use Matlab for example?). Tables 1 and 2 and figures 1 and 2 should be cited in the results section and not in the methods section.
  • Please specify the % you considered to describe the specific phases of gait (lines 81-89)
  • Please describe better data about TD group (number of males?). Were they matched with your sample of children with CP? Please include this in the results section
  • Statistical analysis describes parametric analysis (t-tests). Did you test your dataset for normality? The TD sample is very small and I’m not sure that parametric analysis is the best option. Did you statistically compared children with CP and TD children? Or did you report only qualitative comparison between them? See for example lines 186-187. Please specify. nThis point is also very important e related to results
  • Figures 1 and 2: for GMFCS I and II, I think you reported the mean value of each temporal series. I suggest authors to show mean values as well as the standard deviation of the curves as done for TD subjects.
  • I would support the results section with statistical analysis between children with CP and the TD subjects group. The results as described now are quite confusing because authors compare the three groups but seems they performed a statistical comparison only between groups of CP ( this is what I understood).
  • Results describe Figure 1 and Figure 2 considering a wrong order of joints. For example, figure 2 is described  with panels (a) related to ankle and panels (c) related to hip but it is the opposite. Please correct this.

  • Discussion: this section describes differences between GMFCS I group and GMFCS II group (that are sustained by results) as well as differences between CPs and TDs which are only qualitative assessed in this paper. This makes the paper very weak. Furthermore, the discussion section is very difficult to be understood due to short and not clear sentences (see for example: line 243.  lines 245-247. Line 248 where POWER is included – but no results about joints power are presented)

Minor comments:

  • Lines 57-59 are not clearly related to the whole introduction.
  • Incidents line 81 what do you mean? I would remove
  • Line 109: data were or dataset was
  • Table 1 and table 2 report mean±SD but I see two columns. I imagine that the first one is the mean and the second one the SD but please better specify this (ore use one column writing mean±SD values).
  • Figures are not readable; the labels are very small and difficult to be understood

Author Response

Author´s Reply to the Review Report (jcm-1616553)

Dear Dr. Emmanuel Andrès, dear Prof. Dr. Peter Choong, dear Reviewer,

first of all we want to thank the reviewer for the helpful comments regarding our submission through which we could improve the quality of our manuscript significantly.

The following list includes a detailed overview concerning the comments and the changes we made. The changes are highlighted in yellow color within the manuscript.

Answers to Reviewer 3:

We thank Reviewer 1 for the effort made and appreciate the constructive comments regarding our submission.

We would like to answer the suggestions and comments as follows:

  1. Authors analyzed and reported data about the impaired side of patients included in the analysis. It would have been of interest to include in the report also the unaffected limb and its ability and limitations to execute compensation as suggested by authors in the Introduction section. If this not desired fore some reasons, I suggest to specify in the title and abstract that the paper describe only the impaired side.

Answer: Thank you for your advice as a neutral reader. You are right. The specific paragraph in the introduction was chosen to help the narrative and to outline major differences to bilateral Cerebral Palsy. However, compensatory mechanisms of the uninvolved limb are subject to current investigations. We specified that only the involved side was analyzed in the title and the abstract.

  1. The Materials and methods section has to be restructured: it contains part of results, references to figures and table and does not include important sections such as inclusion and exclusion criteria (that are reported I know but between brackets). Define inclusion criteria before presenting the sample size (that I would move in the results section) (lines 74-79). Consider to use sub-paragraphs to describe 1) the equipment; 2)the population 3) the data analysis (this part is missing and must be added. How did you extracted parameters? Did you use Matlab for example?). Tables 1 and 2 and figures 1 and 2 should be cited in the results section and not in the methods section.

Answer: We would like to thank the Reviewer for this constructive and very helpful suggestion. Accordingly, we restructured the Methods section including subheadings such as “inclusion criteria”, “study population”, “Gait analysis” and “Data analysis” was added. For extraction of the different parameters we used Matlab. The data analysis subparagraph now includes details on data extraction and the use of Matlab. Furthermore, references to Tables and Figures were removed from the Methods sections.

  1. Please specify the % you considered to describe the specific phases of gait (lines 81-89)

Answer: Thank you for your constructive suggestion. According to your suggestions and those of the other reviewers, we performed statistical parametric mapping (SPM) considering the whole gait cycle. As a result, the comparative analyses of specific gait events and gait phases were removed.

  1. Please describe better data about TD group (number of males?). Were they matched with your sample of children with CP? Please include this in the results section

Answer: Thank you for your constructive suggestion. A better and more detailed description of the age-matched TD group (11 males, 17 females; 56 limbs) was included. Consistent with the suggestions of the other reviewers this paragraph was included in the Methods section.

  1. Statistical analysis describes parametric analysis (t-tests). Did you test your dataset for normality? The TD sample is very small and I’m not sure that parametric analysis is the best option. Did you statistically compared children with CP and TD children? Or did you report only qualitative comparison between them? See for example lines 186-187. Please specify. This point is also very important related to results.

Answer: We would like to thank the Reviewer for this constructive and very helpful suggestion. As stated in our answer to comment no. 4, a new (age-matched and bigger) reference group was created and included in SPM (also see answer to comments no. 3). Initially we focused on differences mainly between GMFCS levels. However, in order to improve the statistics of our work and to meet your suggestions, we replaced t-test results shown in Tables 1 and 2 with the SPM-results represented by black bars in the newly designed Figures 1 and 2.

  1. Figures 1 and 2: for GMFCS I and II, I think you reported the mean value of each temporal series. I suggest authors to show mean values as well as the standard deviation of the curves as done for TD subjects.

Answer: Thank you for your constructive suggestion. We added standard deviations to the curves of GMFCS I and II as done for the reference group.

  1. I would support the results section with statistical analysis between children with CP and the TD subjects group. The results as described now are quite confusing because authors compare the three groups but seems they performed a statistical comparison only between groups of CP ( this is what I understood).

Answer: As suggested, the TD group has been included in the newly performed SPM analysis. Changes have been done throughout the whole manuscript accordingly (also see answers to comments no. 3 and 5).

  1. Results describe Figure 1 and Figure 2 considering a wrong order of joints. For example, figure 2 is described with panels (a) related to ankle and panels (c) related to hip but it is the opposite. Please correct this.

Answer: Thank you for your advice as a neutral reader. In the newly designed Figures the order of panels was corrected.

  1. This section [Discussion] describes differences between GMFCS I group and GMFCS II group (that are sustained by results) as well as differences between CPs and TDs which are only qualitative assessed in this paper. This makes the paper very weak. Furthermore, the discussion section is very difficult to be understood due to short and not clear sentences (see for example: line 243. lines 245-247. Line 248 where POWER is included – but no results about joints power are presented)

Answer: We would like to thank the Reviewer for this constructive and very helpful suggestion. Indeed, comparative statistics was performed only for GMFCS I vs. GMFCS II. Differences between this specific subgroups and the TDs were described qualitatively. However, as you suggested in comment no. 7 and consistent with the suggestions of the other Reviewers, we performed SPM between all three groups. As a result, changes concerning the Methods and Results sections as well as the Discussion were done. Moreover, comprehensibility of the Discussion section was improved, particularly with respect to the given example.

  1. Lines 57-59 are not clearly related to the whole introduction.

Answer: You are right. This paragraph was written to help the narrative and to outline major differences to bilateral Cerebral Palsy. Due to the naturally asymmetry of the disorder gait pathologies in unilateral Cerebral Palsy are more complex. However, compensatory mechanisms of the uninvolved limb are subject to current investigations. The specific paragraph has been removed.

  1. Incidents line 81 what do you mean? I would remove

Answer: We meant gait events. However, due to performing SPM over the whole gailt cycle, the entire paragraph concerning gait events and gait phases was removed.

  1. Line 109: data were or dataset was

Answer: Was changed to “data were”.

  1. Table 1 and table 2 report mean±SD but I see two columns. I imagine that the first one is the mean and the second one the SD but please better specify this (ore use one column writing mean±SD values).

Answer: According to the suggestions of all reviewers, we performed SPM as a continuous data analysis for the whole gait cycle. As a result, Tables 1 and 2 were removed. SPM results represented by black bars were integrated in the newly designed Figures 1 and 2.

  1. Figures are not readable; the labels are very small and difficult to be understood

Answer: Thank you for your advice as a neutral reader. Accordingly, we adjusted the size of the labels to improve readability. We hope that the newly created figures now meet your requirements.

We hope that our manuscript now, after profound revision, accomplishes the high standards of Journal of Clinical Medicine.

We are looking forward to hearing from you.

Yours sincerely,

Stefanos Tsitlakidis                              and                  Marco Götze

(Submitting author)                                                     (Corresponding author)

Round 2

Reviewer 2 Report

The manuscript has improved greatly by the implementation of SPM as a statistical tool. The authors answered my question, but partly this was not sufficient in my perspective, or I am wondering why the information was not added to the manuscript. Therefore, I am going to repeat some of the questions.

Comment: “At least, it would be great to have a number of different gait patterns present in both groups.” This was ignored and the information was not added to the manuscript.

Why did you not add the sampling frequency (1080 Hz) for your force plates to your manuscript?

Comment about the paper of Kadaba et al. (1990): I want to clarify that I don’t have a problem with this publication being used as a reference to the PiG model! However, usually a segment has to be defined by at least three markers to allow for calculations in all three planes, which concerns your presented frontal and transversal kinematic and kinetic values at the ankle. In my opinion, if presenting the results, at least the limitation of the calculations should be mentioned.

How were you your ankle and knee axis defined? I.e. knee and ankle axis definition influence hip and knee rotation.

Please state that you used raw data in your manuscript. This is uncommon, usually raw force plate and trajectory data is somehow filtered before joint kinematics and kinetics are calculated.

I asked you “Please also mention if your joint moments presented are external or internal moments.”. This was not done yet. Please add this information to your manuscript.

Comment about the foot progression angle: you do not report ankle rotation (kinematics) but you present the ankle rotation moment (kinetics). Instead of ankle rotation you present the foot progression angle. I agree that within the use of PiG the presentation of foot progression rather than the ankle rotation is common. But if presenting the joint moments showing the corresponding kinematics as well could be advantageous. Or at least mentioning your response ““Foot progression” as long foot axis in relation to the gait direction. “High” or positive values being intoeing.” In the manuscript could be useful for the reader.

Author Response

Author´s Reply to the Review Report (jcm-1616553)

Dear Dr. Emmanuel Andrès, dear Prof. Dr. Peter Choong, dear Reviewer,

first of all we want to thank the reviewer for the helpful comments regarding our submission through which we could improve the quality of our manuscript significantly.

The following list includes a detailed overview concerning the comments and the changes we made. The changes are highlighted in yellow color within the manuscript.

Answers to Reviewer 2:

We thank Reviewer 1 for the effort made and appreciate the constructive comments regarding our submission.

We would like to answer the suggestions and comments as follows:

  1. “At least, it would be great to have a number of different gait patterns present in both groups.” This was ignored and the information was not added to the manuscript.

Answer: Our apologies, we misinterpreted it as part of the discussions during the review process. Additional information on the distribution of gait patterns within the GMFCS level I and level II groups was added in the Methods section/subheading Study Population.

  1. Why did you not add the sampling frequency (1080 Hz) for your force plates to your manuscript?

Answer: We would like to apologize for not adding this information earlier. We did not want to give the impression that we ignored your comment. We now specified the sampling frequency of the force plates in the Methods section.

  1. Comment about the paper of Kadaba et al. (1990): I want to clarify that I don’t have a problem with this publication being used as a reference to the PiG model! However, usually a segment has to be defined by at least three markers to allow for calculations in all three planes, which concerns your presented frontal and transversal kinematic and kinetic values at the ankle. In my opinion, if presenting the results, at least the limitation of the calculations should be mentioned.

Answer: The “Plug-in Gait model” is in fact a software code developed by Vicon. It largely refers to Kadaba et al. (1990). However, throughout the years, details in the procedure with patients and also within the software have changed without ever having been documented by proper scientific publication. Nevertheless, this is the standard in most clinically applied gait studies. We added the reference by Baker et al. where these details are mentioned.

Along the changes of the software, also the foot, originally defined only as a 2D-segment (i.e. a line from ankle center to toe marker), has later been defined as a 3D-segment by including the heel marker. This in principle then allows for full 3D joint kinematics and kinetics and meanwhile is accepted standard.

  1. How were you your ankle and knee axis defined? I.e. knee and ankle axis definition influence hip and knee rotation.

Answer: We added a sentence about the knee axis determination via a knee alignment device since different clinical protocols exist here. The ankle joint axis is determined via Plug-In Gait exactly as described in the original article by Kadaba et al. (1990). Trunk kinematics was calculated via other procedures. Reference is given.

  1. Please state that you used raw data in your manuscript. This is uncommon, usually raw force plate and trajectory data is somehow filtered before joint kinematics and kinetics are calculated.

Answer: There must be a misunderstanding. According to the Plug-In Gait procedures a quintic spline filter based on code written by Herman Woltring is applied to marker trajectory data as well as force plate data. Since these are common procedures within Plug-In Gait, we prefer not to burden the manuscript with these technical details.

  1. I asked you “Please also mention if your joint moments presented are external or internal moments.”. This was not done yet. Please add this information to your manuscript

Answer: We added this information in the methods. Internal moments are reported.

  1. Comment about the foot progression angle: you do not report ankle rotation (kinematics) but you present the ankle rotation moment (kinetics). Instead of ankle rotation you present the foot progression angle. I agree that within the use of PiG the presentation of foot progression rather than the ankle rotation is common. But if presenting the joint moments showing the corresponding kinematics as well could be advantageous. Or at least mentioning your response ““Foot progression” as long foot axis in relation to the gait direction. “High” or positive values being intoeing.” In the manuscript could be useful for the reader.

Answer: We added this sentence in the methods: “Foot progression, describing the orientation of the foot’s long axis in relation to the gait direction has been chosen instead of ankle rotation, since clinically this parameter is given more importance.” Positive values report intoeing but this should be clear in the figure where an explicit axis description is given.

We hope that our manuscript now, after profound revision, accomplishes the high standards of Journal of Clinical Medicine.

We are looking forward to hearing from you.

Yours sincerely,

Stefanos Tsitlakidis                          and                                  Marco Götze

(First author)                                                            (Corresponding author)

Reviewer 3 Report

I would like to thank the authors for their work. I still have some concerns about their manuscript and I report below my comments.

  1. Line 85: IGA acronym is presented before authors define it (line 91). Please correct this
  2. Line 96: the data WERE extracted. Please correct it.
  3. Line 100: is the Matlab version correct?I see many “4”
  4. Please better explain in the methods section the one dimensional statistical parametric mapping (SPM) to make your study repeatable. The analysis is of interest for other clinicians/researchers and the procedure you used should be described.
  5. Paragraph 3.1 is now improved. However, each joint is named with a letter (a, b, .., e) but 3 plots are showed for each line. It would be better to name each plot with a letter, thus supporting the readability of the results. This comment has to be considered for Figure 1 as well as Figure 2.
  6. In my previous revision I asked authors to “specify the % you considered to describe the specific phases of gait”. Authors stated that they did not include the definition of % of gait phases because they performed SPM analysis in the revised version of the manuscript. However, authors refer to gait phases in the results description. As a researcher expert in gait analysis, I can follow the results. However I believe that non expert could have difficulties in understanding your results. Therefore, I would suggest to include the definition of the % of the gait cycle to improve readability
  7. Authors refers to subjects with CP as “both subgroups” in the results section. It may be difficult to understand. Why TD are a group and GMFCS I CP a subgroup? Please revise the text accordingly
  8. The caption of Figure 1 and 2 should explain what are the straight lines (mean of impaired limb of subjects with CP) and the dotted lines (+/- 1 SD ?).
  9. Lines 124-125: “For descriptive statistics the mean and the standard deviation (SD), the minimum, the maximum and the range were calculated and displayed graphically.” I do not see the minimum, maximum and range. Please correct it
  10. Line 123 a dot is placed in an incorrect position. Please remove it

Author Response

Author´s Reply to the Review Report (jcm-1616553)

Dear Dr. Emmanuel Andrès, dear Prof. Dr. Peter Choong, dear Reviewer,

first of all we want to thank the reviewer for the helpful comments regarding our submission through which we could improve the quality of our manuscript significantly.

The following list includes a detailed overview concerning the comments and the changes we made. The changes are highlighted in yellow color within the manuscript.

Answers to Reviewer 3:

We thank Reviewer 1 for the effort made and appreciate the constructive comments regarding our submission.

We would like to answer the suggestions and comments as follows:

  1. Line 85: IGA acronym is presented before authors define it (line 91). Please correct this.

Answer: You are right. We changed that accordingly.

  1. Line 96: the data WERE extracted. Please correct it.

Answer: Has been corrected accordingly.

  1. Line 100: is the Matlab version correct?I see many “4”

Answer: Has been changed accordingly.

  1. Please better explain in the methods section the one dimensional statistical parametric mapping (SPM) to make your study repeatable. The analysis is of interest for other clinicians/researchers and the procedure you used should be described.

Answer: We added information on this in the Methods section. References are given.

  1. Paragraph 3.1 is now improved. However, each joint is named with a letter (a, b, .., e) but 3 plots are showed for each line. It would be better to name each plot with a letter, thus supporting the readability of the results. This comment has to be considered for Figure 1 as well as Figure 2.

Answer: Figures 1 and 2 have been improved accordingly.

  1. In my previous revision I asked authors to “specify the % you considered to describe the specific phases of gait”. Authors stated that they did not include the definition of % of gait phases because they performed SPM analysis in the revised version of the manuscript. However, authors refer to gait phases in the results description. As a researcher expert in gait analysis, I can follow the results. However I believe that non expert could have difficulties in understanding your results. Therefore, I would suggest to include the definition of the % of the gait cycle to improve readability.

Answer: Our apologies for any misunderstanding due to the change in analysis. We refrained from analyzing specific sub-phases of the gait cycle and replaced this now by SPM analysis. We hope to have clarified this in the manuscript accordingly. However, in the results and discussion section, the periods of significance within the gait cycle describing differences between groups, have been characterized by typical clinical terms naming sub-phases of gait. A definition of gait phases was included in the methods section.

  1. Authors refers to subjects with CP as “both subgroups” in the results section. It may be difficult to understand. Why TD are a group and GMFCS I CP a subgroup? Please revise the text accordingly.

Answer: The text has been revised throughout the whole manuscript.

  1. The caption of Figure 1 and 2 should explain what are the straight lines (mean of impaired limb of subjects with CP) and the dotted lines (+/-1SD?).

Answer: Thank you for your constructive suggestion. You are right, this is missing. We adjusted the captions of the figures with improved description.

  1. Lines 124-125: “For descriptive statistics the mean and the standard deviation (SD), the minimum, the maximum and the range were calculated and displayed graphically.” I do not see the minimum, maximum and range. Please correct it.

Answer: Thank you for your advice as a neutral reader. We have corrected this sentence accordingly.

  1. Line 123 a dot is placed in an incorrect position. Please remove it.

Answer: The dot has been removed.

We hope that our manuscript now, after profound revision, accomplishes the high standards of Journal of Clinical Medicine.

We are looking forward to hearing from you.

Yours sincerely,

Stefanos Tsitlakidis                       and                                 Marco Götze

(First author)                                                        (Corresponding author)
